# Social Cognition and Prosocial Behavior in Children with Attention Deficit Hyperactivity Disorder: A Systematic Review

**DOI:** 10.3390/healthcare11101366

**Published:** 2023-05-10

**Authors:** Olber Eduardo Arango-Tobón, Alexandra Guevara Solórzano, Silvia Juliana Orejarena Serrano, Antonio Olivera-La Rosa

**Affiliations:** 1Basic and Applied Neuroscience, Faculty of Psychology, Luis Amigó Catholic University, Medellin 680001, Colombia; alexandraguevara.psi@gmail.com (A.G.S.); antonio.oliverade@amigo.edu.co (A.O.-L.R.); 2Basic and Applied Neuroscience, School of Medicine, Industrial University of Santander, Bucaramanga 680002, Colombia; silvia.orejarena@correo.uis.edu.co

**Keywords:** ADHD, prosocial behavior, social cognition, systematic review, theory of mind

## Abstract

The purpose of this research is to analyze the empirical evidence on the relationship between social cognition and prosocial behavior in children and adolescents with Attention Deficit Hyperactivity Disorder (ADHD). A systematic review was carried out following the PRISMA guidelines of empirical studies found in PubMed and Scopus databases, including a total of 51 research studies. The results indicate that children and adolescents with ADHD have deficits in social cognition and prosocial behavior. For children with ADHD, their deficits in social cognition highlight their difficulty in the process of theory of mind, emotional self-regulation, emotion recognition and empathy, affecting prosocial behavior, evidencing difficulty in personal relationships, and the creation of emotional bonds with their peers.

## 1. Introduction

It is currently established that the way in which humans relate to others consists of achieving success in terms of social recognition; thus, identifying the appropriate forms of interaction generates debates that bring modern studies to focus on social cognition, as a neurobiological, psychological, and social process, through which the subject perceives, recognizes, and evaluates social events, in order to build a representation of what happens between interactions, and find the appropriate response to each situation by evaluating the intentions of others, based on language and behavior [1].

There are skills that are linked to social cognition, such as the ability to perceive the emotions of others, identifying facial signals, posture and prosody in others, as well as reasoning regarding mental states (theory of mind—ToM), empathy, and regulation of the emotions of others [2].

One of the population groups most affected in terms of social cognition are individuals with Attention Deficit Hyperactivity Disorder (ADHD), defined as a neurodevelopmental disorder identified by problematic levels of inattention and/or hyperactivity—impulsivity. Inattention implies the inability to follow tasks and lose objects at levels that are incompatible with age and developmental level, while hyperactivity/impulsivity implies excessive activity for age and developmental level, which interferes in the academic, personal, and social spheres of those diagnosed [3].

Neurocognitive deficits in ADHD such as disinhibition, inattention, impulsivity, and hyperactivity have been linked to deficits in social behavior and dimensions of social cognition, such as ToM, emotional processing, empathy, and prosocial behaviors. Children with ADHD are often negligent in identifying socially relevant cues that generate socially inappropriate responses with their peers, teachers, and parents, causing alterations in their adaptive behavior [4,5].

Social cognition has different dimensions and perspectives from which it can be understood. However, there is a consensus that refers to social cognition as an ability possessed by some species of social animals, including man, which allows them to recognize, manipulate, and behave with respect to information that is socially relevant to them. It is also well established that this ability requires a neural system to carry out the processing of social cues linked to motivational and emotional processes and adaptive behaviors based on the construction of representations of relationships between oneself and others and the flexible use of such representations to guide social behavior [6].

Several terms used to refer to social cognition have been identified in the scientific literature. Empathy has been used in the context of social cognition to designate the abilities to recognize emotional states in others and to relate to the feelings of others. On the other hand, the theory of mind is frequently used to emphasize the idea that we have a metacognitive understanding of our own minds and those of others. Emotional processing involves the way in which each one of us cognitively and effective elaborates the representations of our emotions (emotional awareness) and that of others.

Emotional processing also involves the implicit attribution that guides the anticipation of emotional and behavioral patterns (e.g., the detection and perception of facial and body expressions). All these components of social cognition are closely linked to the regulation of social intentions, the prediction of the behavior of others, and the ability to adapt to the demands of the context [7,8].

There is empirical evidence on the relationship between social cognition and prosocial behavior in children and adolescents with ADHD; some systematic reviews and meta-analyses on the dimensions of social cognition have been found. Ref. [9] analyzed deficits in the ToM and empathy with an emphasis on affected neuronal substrates [2]. Ref. [10] focused on the analysis of emotion recognition and the ToM, comparing children with ADHD and ASD; Ref. [11] analyzed only the recognition of facial emotions; Ref. [5] reviewed the relationships between executive functions and ToM [5]; Ref. [12] conducted a systematic review with clinical trials on the effects of pharmacological treatments on empathy and social cognition in children and adolescents with ADHD.

This systematic review extends the aforementioned literature by addressing the relevant dimensions of social cognition (ToM, emotional awareness, emotion recognition, and empathy) and its relationship with prosocial behavior in children and adolescents with ADHD (ToM, emotional awareness, emotion recognition, and empathy). This is of great importance for all those researchers who are interested in the subject, since there seems to be a consensus about deficits in social cognition and prosocial behaviors in children and adolescents with ADHD.

This information could generate a call for attention to clinical and educational personnel to take into account in the evaluation not only cognitive, behavioral, and neuropsychological symptoms, but also measurements of components in social cognition. Thus, measuring these components could shed light on more comprehensive interventions that involve aspects such as empathy, ToM, emotional awareness, and prosocial behaviors in this population.

This systematic review aims to describe the evidence on the relationship between social cognition and prosocial behavior in ADHD in children and adolescents, providing answers for the interaction processes and providing information on the importance of social cognition in human integrality.

Some studies [2,9,13,14] show that children with ADHD exhibit fewer social cognition skills than children without ADHD, evidencing low levels of empathy, theory of mind, emotional self-regulation, and the interpretation of others’ thoughts and emotions, causing limitations in the creation of social bonds and altering the way in which they respond to certain external stimuli.

The objective of this systematic review is to analyze the empirical evidence on the relationship between social cognition and prosocial behavior in children and adolescents with ADHD.

## 2. Method

A systematic review of the scientific literature published on social cognition, prosocial behavior, and ADHD was carried out. The PRISMA [15] (Moher et al., 2009) protocol for systematic reviews was followed. The following sections describe in detail the elaboration process in its phases.

### 2.1. Initial Search

The first searches were performed in February 2021 by combining the terms “social cognition” and “ADHD”, and using the “AND” connector in Google Scholar, PubMed, and Scopus databases. These searches yielded 91,620 research studies, which provide a current overview of the topic to be discussed. Google Scholar was the database that produced the most records (*n* = 89,332). We made the decision not to analyze these results because this database collects a lot of information without quality control from the sources and a wide variety of publications such as books, book chapters, communications, and presentations of scientific events, technical reports, theses or degree works, non-scientific works from repositories of different institutions, teachers’ subject guides, PowerPoint presentations, and Word documents, among others, without a filter that categorizes the quality of these products with published research articles in scientific databases.

### 2.2. Systematic Search

The systematic search was performed again in April 2021 in PubMed and Scopus, delimiting the results from the year 2000.

The search combinations used to determine the articles that were reviewed were typed into the two databases as follows: “Social Cognition” AND “ADHD”, “Empathy” AND “ADHD”, “Theory of mind” AND “ADHD”, “Emotional regulation” AND “ADHD”, “Social perception” AND “ADHD” “Social inference” AND “ADHD”, “Emotional awareness” AND “ADHD” and “Regulation of social interaction” AND “ADHD”; “Social cognition” AND “ADHD”, “Empathy” AND “ADHD”, “Theory of mind” AND “ADHD”, “Emotional regulation” AND “ADHD”, “Social perception” AND “ADHD”, “Social inference” AND “ADHD”, “Emotional awareness” AND “ADHD” and “Regulation of social interaction” AND “ADHD” were also searched for combinations. In each search, different results were obtained, showing little approach to the topic in concepts such as “social inference”, “emotional awareness”, and “Regulation of social interaction”, but greater conceptual openness is shown in topics such as “social cognition”, “ToM”, “empathy”, and “emotional regulation”, all in combinations with “ADHD”. Regarding the results obtained with the key concepts in the Spanish language, there was not enough research to be considered for conducting the systematic review. Specifically, 440 results were obtained in PubMed and 1848 in Scopus. Before proceeding to the selection of articles, the inclusion and exclusion criteria are defined.

### 2.3. Inclusion Criteria

To only review empirical research on whether or not there is a deficit in social cognition and/or prosocial behavior in children with ADHD. Only empirical articles were included since they have a correlational or experimental scope to provide evidence on the relationship between dimensions of social cognition and prosocial behaviors in samples of children and adolescents with ADHD.

Studies that included children and adolescents with ADHD as units of analysis. Only studies with children and adolescents were included because the symptomatology is similar in both groups and very different from that of adults with ADHD. In addition, the trajectory of symptoms over time and the characteristics of dimensions of social cognition and prosocial behaviors have been better studied in children and adolescents than in adults.

Research conducted between 2000 and 2021. Articles published before the year 2000 were not included because the databases consulted did not find records prior to the year 2000 that related to the objective of this systematic review and at the beginning of the year 2022, the systematic search began, for which the search window was from the year 2000 to 2022.

The articles are aimed at the study of “social cognition”, “prosocial behavior”, and “ADHD”. These topics are of current interest and show studies that, although they contribute to the understanding of the relationships between these variables in children and adolescents with ADHD, the evidence is still inconclusive and sometimes becomes contradictory.

Articles written in English and Spanish. Only two articles in Spanish were found and included in this review. Other languages were not included in this review because no articles were found that would contribute to the objective of this systematic review.

### 2.4. Exclusion Criteria

Research prior to the year 2000 was not addressed.

Articles studying social cognition, prosocial behavior, and ADHD in adults.

Articles involving biological research, qualitative research, and theoretical and systematic reviews were excluded.

According to the inclusion and exclusion criteria, and the titles, 181 articles were considered suitable (after eliminating 26 duplicates between the two databases and 23 duplicates in the Scopus database). Subsequently, the abstract of each of the articles was read, finally selecting 51 research studies that fully met the established criteria. The 130 excluded articles focused on biological processes of the sample, had little depth in terms of social cognition, prosocial behavior, and ADHD, and were qualitative research or theoretical or systematic reviews.

All of the selected research named at least one of the systems of social cognition and/or prosocial behavior and related it to ADHD in children and adolescents, highlighting conclusions about their relationship and importance in the social development of the sample participants. Some articles also created relationships with the positive effects of methylphenidate (MPH) on social cognition processes in children and adolescents with ADHD; the decision was made to include them because their approaches were in line with the other research studies conducted and their conclusions were comparable to the other studies.

In most of the research studies, they evaluated theory of mind and emotional regulation; empathy was also important in this review; the other aspects of social cognition identified were social perspective-taking and pragmatic language, but they were not as repetitive in the different research studies (Figure 1).

## 3. Results

This section discusses the analysis of the research, providing a greater understanding and integration of the results obtained. A summary of the results of the selected studies is provided in Table 1.

According to the total samples of the reviewed studies (*n* = 2439), 96.4% presented deficits in social cognition. This indicates that the diagnostic condition of the participants is closely related to deficits in social cognition and limitations in prosocial behaviors. The results indicate that the theory of mind (ToM) is the dimension affected most in children and adolescents with ADHD; however, emotional awareness, emotion recognition, and empathy are also deficits that generate a link with prosocial behavior, evidencing implications that affect the social development of children with ADHD.

### 3.1. Theory of Mind

Some authors, using Happé’s Strange Stories, belief recognition instruments, and the NEPSY II, did not find significant differences between the ADHD and control groups, showing, according to the authors, that because of the way the indication is given, the tests used and the type of ADHD can lead to unreliable results [16,17,18]. Other studies using the Happé’s Strange Stories Test, the ice cream task, and the Sally Anne and the Marble task confirmed that the ADHD group showed problems in reading thoughts and feelings in others compared to the control group [19,20,21].

It is important to note that authors reported that ToM deficits play an important role in the impairment of social cognition and peer relationships in children with ADHD [22]; the same author confirmed his first hypothesis, adding the importance of pharmacological treatment with methylphenidate for improvements in ToM in children with ADHD [23]. They also stated that attentional difficulties are responsible for ToM deficits in ADHD, contributing to the negative involvement of emotional and social processes in this population [24].

The deficit in ToM is demonstrated by inferior performance in the comprehension of intentional behaviors in children with ADHD [25], a conclusion shared by other authors, who stated that children with ADHD have difficulties in inferring and understanding the actions of others based on their beliefs, affecting the mentalization processes of ToM [26]. Others reported that ToM dysfunction is not a primary deficit in children with ADHD; it is executive problems that impede the representation of mental states and ToM skills [27,28].

The influence of ToM skills on emotional regulation skills is shown within the ADHD group [29]. In addition, it was concluded that children with ADHD performed worse on measures of cognitive ToM and affective recognition, but on affective ToM they had the same skills as children in the control group [30]. The impairments in ToM cause difficulties in communication and social interaction [31]. These results refer that the deficient functioning of ToM in children with ADHD [32,33,34] is associated with a lower socioemotional functioning, although it improves with age; however, it is not positively or negatively affected by the administration of methylphenidate [35] (Golubchik and Weizman, 2020).

Emotional processing (emotional self-regulation, emotional awareness, and recognition of emotions).

Some reviewed studies exposed the deficits in emotional self-regulation in children with ADHD. Following the application of the Emotional Intensity Scale for Children (EISC), the authors pointed out that children with ADHD exhibited a higher emotional intensity response compared to the control group [36]; they also evaluated emotional self-regulation using the Emotional Regulation Task, concluding that there is a difference in the processes of emotional self-regulation between children with and without ADHD; however, the responses of children with inattentive ADHD were mostly similar to those of children in the control group [13].

On researching emotional awareness, applying the Emotional Intensity Scales, the Contingency Scale: main version (External Contingency Scale), and the Event Scale (Immediate/Future Emotions Scale, the results indicated differences in self-regulation between children with and without ADHD, especially when facing negative events [37] (Jesen and Rosen, 2004). They evaluated the processing of emotions using the Minnesota and Spencer emotional processing test and deepened this dimension by evaluating the Child Behavior Checklist (CBCL). Both research studies confirmed differences between the ADHD and control groups, in addition to defining that emotional dysregulation in children with ADHD is associated with elevated rates of anxiety and behavioral disorders [21].

It is defined that children with and without ADHD differ in the application of problem-oriented emotional regulation strategies [38]. Poor emotional awareness is also reported to be related to reactive behaviors in children with ADHD and is positively associated with comorbid externalizing disorders, which is indicative of dysregulated emotional reactivity rather than planned and intentional misbehavior in children with ADHD [39].

The researchers determined that children with ADHD show weaker coherence between emotion, reactivity, and regulation indices compared to children without ADHD [40]. In turn, after evaluating with the NEPSY-II, they reported that there were no significant differences between children with and without ADHD in terms of emotion regulation [18]. They also found differences in emotional regulation between children with and without ADHD; however, they specify that this regulation is more cognitive than emotional [41]. Other authors supported the idea that children with ADHD who had deficits in the ability to reason about the emotions of others have difficulty expressing and regulating their emotions [29,35,42].

Using the Child Behavior Checklist (CBCL) and the Leiter International Performance Scale, it was determined that children with ADHD show worse behavior when they are involved in group situations without or with minimal adult supervision, in addition to identifying that the presence of emotional dysregulation among preschoolers with ADHD is associated with considerable levels of behavioral comorbidity and social dysfunction [43]. Other studies also agreed with the differences between groups and add that adolescents with ADHD showed poorer performance in emotion regulation, and also reported that difficulties in emotional regulation correlate with the severity of ADHD, which influences the development of good interpersonal relationships [44,45].

Among the dimensions of social cognition, we found the recognition of emotions, identifying that only one research study, which evaluated participants using the Emotion Recognition Task (ERT), concluded that there were no significant differences between the two groups, obtaining a result that the ADHD group was more confident in the recognition of sad and angry faces compared to the control group, in addition to not differing in the accuracy of emotion recognition [46].

The results of the remaining research that studied emotion recognition showed social cognition deficits in the ADHD group. Emotion recognition was assessed using the Emotion Behavior Checklist (EEC), finding that children with ADHD have difficulty interpreting and describing the emotions of others [36]. There is also evidence that children with ADHD demonstrate mild to moderate deficits in their ability to perceive facial expression and prosody [47]. The studies showed that the ADHD group exhibited a general deficit in the decoding of emotional facial expressions, with specific deficits in the identification of anger and sadness [48], as well as difficulties in understanding the links of emotions expressed by others and in identifying the situation in which these emotions occur [49].

In another study, it was appliedthe Emotion Recognition Task to their participants, concluding that children with ADHD exhibit deficits in emotion processing, affecting the recognition of facial emotions from contextual information [50]. Through the application of the Emotion Evaluation Test (EET), another study showed that children with ADHD exhibit a deficit in the recognition of negative and positive emotions [51,52].

By means of an animated transformation task (Radboud Faces Database) [53] and Software Neurobehavioral System, they achieve lower scores in the emotional categories of children with ADHD [54] (Jusyte et al., 2017) and confirm the difference between the control group and ADHD group in the recognition of angry expressions [55]. Facial and auditory emotion recognition problems are confirmed to be characteristic of ADHD; however, children with ADHD did not show emotion recognition deficits as strong as adolescents with ADHD, which gives the impression that emotion recognition deficits worsen from childhood to adolescence [30,56].

### 3.2. Empathy

Another dimension of social cognition is empathy, which was studied by means of the Empathy Task and the Empathy Response Task (ERT), determining that children with ADHD are less likely to be empathetic to the emotions of storybook characters compared to the control group [36]. It was confirmed that children with ADHD were rated as less empathetic by their parents; however, due to social desirability in self-reporting, the difference was not so distant between the experimental group and the control group [20,57]. In another study, they applied the Emotional Reactivity Index (ERI), indicating deficits in children with ADHD and stating that the linkage to pharmacological treatment with methylphenidate can improve empathy processes [22].

Lower levels of empathy are found in children with ADHD versus feelings of sadness in others [58], but after a pharmacological treatment of more than twelve weeks, the capacity to be empathic increased [51], which could equalize the empathic processes of children with ADHD with the results of the control group [23]. In turn, the Interpersonal Reactivity Index (IRI) was applied, and it showed that cognitive empathy is affected in children with ADHD; however, differences were not significant in terms of emotional empathy [30].

Empathy processes are fundamental for interpersonal relationships and the bonds that develop from them, which is why, when there are deficiencies in empathy, there are difficulties in socializing and maintaining good relationships, directly affecting adaptation to the environment and not having adequate prosocial behavior to respond to it. Therefore, this systematic review also aims to find deficiencies in the prosocial behavior of children and adolescents with ADHD.

### 3.3. Prosocial Behavior

After reviewing the research on prosocial behavior, all the studies agree that it impacts groups of children with ADHD. The results show that children with ADHD show deficits in social performance in relation to controls; however, the main characteristics of these deficits differ between types of ADHD. For example, children with inattentive ADHD were rated as having more passive behaviors, while children with combined ADHD showed greater social aggressiveness, with passiveness and aggressiveness being the factors that influenced their prosocial behavior [13]. Other research studies assessed prosocial behavior through the social goals interview [59], finding that the control group had better results in the social aspect [16,60]. Through the Interpersonal Negotiation Strategies Interview, it was found that the skills of the ADHD group were less developed than those of the control group [57].

In another study they used the Child and Adolescent Scale of Participation (CASP) instrument, from which they concluded that children with ADHD exhibit problems of social competence, experiencing social rejection, as well as secondary difficulties with mood. They added to their results that the combined ADHD and inattentive ADHD groups showed significant differences in the interpretation of emotional and nonverbal cues in a direct measure of social perception compared to controls. These findings show a link between inattention and social perception that is separate from impulsivity difficulties [61].

Social behavior was evaluated using the Impairment Rating Scale (IRS) and the methodology of toys that simulate a situation, finding that adolescents with ADHD showed deficiencies in social comprehension and problem-solving skills compared to adolescents without ADHD [62]. In turn, other studies recognized a deficit in prosocial behavior in children with ADHD, showing greater difficulty in establishing relationships and responding to obvious social cues [52,58,63,64]. They researched social inference and found that children with ADHD showed deficits in understanding paradoxical sarcasm as well as being less accurate in basing their decision on the speaker’s feelings compared to the control group [27,28,65].

To conclude, the empirical evidence shows that children and adolescents with ADHD exhibit deficits in social cognition, especially in the dimensions of ToM, emotional awareness, emotion recognition, and empathy; these dimensions are the starting point for the prosocial behavior of the ADHD population to exhibit difficulties in creating adequate bonds in addition to responding appropriately to the signals established in the social environment.

## 4. Discussion

The aim of this systematic review was to establish the empirical evidence on the relationship between social cognition, prosocial behavior, and ADHD in children and adolescents, in order to verify whether the evidence shows that social cognition and/or prosocial behavior are affected in children with ADHD.

The evidence on the relationship between social cognition and prosocial behavior in children and adolescents with ADHD is mostly consistent, and indicates deficits in social cognition and prosocial behavior in this neurodevelopmental disorder. The results of these investigations point to neurocognitive deficits that include emotional processing, empathy, and ToM and conclude that these alterations have a direct impact on social behavior, which causes these children to exhibit negligent responses to the changes they must make in their adaptive behavior and ignore cues. Social skills are important for interactions with peers, teachers, and caregivers.

### 4.1. Theory of Mind and ADHD

The ToM was studied in 24 articles, and it is among the dimensions of social cognition that are affected in children with ADHD. Through instruments such as Happé’s Strange Stories, the Reading the Mind in the Eyes Test, and the Faux Pas Recognition Task (FPR), among others, it is identified that the ToM is affected in children with ADHD, causing difficulties in reading the thoughts and feelings of others; they also exhibit trouble interpreting the language of others, which generates difficulties with peers. Items that did not show deficits in social cognition were assessed via the NEPSY II, Appearance-Reality (ToM1), and the unexpected location (ToM2) task was modeled after Sally and Anne’s Task and unexpected content (ToM3).

In turn, this review finds that children with ADHD respond with the same accuracy to questions that involve the beliefs of others and that the type of ADHD is fundamental to demonstrating participants’ deficits, influencing the children’s answers, and leading to low performance in the understanding of intentional behaviors.

Ref. [26] justify that the way in which the instructions are socialized at the time of applying the test, and the tools used, whether auditory or verbal, facilitate the development of the items, generating a non-significant difference between the evaluated groups. Other research on ToM concludes difficulties in inferring and understanding the actions of others based on their beliefs. They report that the ADHD population has a lower ToM in the cognitive dimension than the control group; however, the ToM In its emotional dimension has the same abilities as the control group [30].

It is considered that scores on instruments assessing ToM may improve so that the results resemble those of the control group after 12 weeks of treatment with methylphenidate [23]; however, they report that the improvement in ToM develops throughout the life cycle, without the influence of methylphenidate. ToM functioning in children with ADHD is related to a poor capacity for social behavior with peers [35].

### 4.2. Emotional Processing

One of the dimensions of social cognition affected in ADHD is emotional self-regulation, where it is shown that children and adolescents with ADHD exhibit a more intense response compared to the control groups. They report that children with inattentive ADHD have intact emotional regulation skills, unlike those with hyperactive and combined ADHD who are characterized by more reactive behaviors, preventing them from controlling situations that generate emotional responses [13].

The emotional response of children with ADHD is more intense, especially when facing negative events [37]. It is also confirmed that there is greater emotional dysregulation in patients with ADHD, due to the high rates of anxiety and behavioral disorders; these comorbidities and the specific characteristics of ADHD (inattention and hyperactivity) define the difficulty in terms of emotional functioning [66].

To assess emotional dysregulation in ADHD, several studies applied the Questionnaire on Emotion Regulation in Children and Adolescents (FEEL-KJ) and the Emotion Regulation Checklist (ERC); these instruments contain self-reports, which revealed differences in emotional regulation, confirming the deficit in emotional awareness in children and adolescents with ADHD. There are externalization problems [39], that indicate dysregulated emotional reactivity, which shows the difficulty of generating planned and intentional behaviors in situations to which ADHD children are exposed, showing a weaker coherence between the indices of emotion, reactivity, and regulation [40].

Deficiencies in patients with ADHD regarding emotion are based on their poor ability to reason about the emotions of others, which causes difficulties in expressing and regulating their own emotions; therefore, children with ADHD show worse emotional behavior when they do not have adult supervision and are exposed to reactive situations. The effects on this dimension of social cognition cause significant impacts on the interpersonal relationships of diagnosed participants, associating their emotional dysregulation with social dysfunction.

The recognition of emotions in others is impaired in the ADHD population, which indicates a lower likelihood of being empathic with others and also makes it difficult to interpret and describe other people’s feelings [36]. This is also related to their deficit in perceiving facial expressions and prosody, identifying greater difficulties in decoding anger and sadness in the faces of others [48].

The population with ADHD exhibits obstacles in processing emotional information; therefore, they do not correctly understand the links between emotions, nor are they able to identify the situations in which different emotions occur [49]. There is a general deficit in the processing of emotions and not being able to focus on contextual information, which prevents the interpretation and proper use of strategies for relating to others. Additionally, more difficulty is exhibited in adolescents than in children, determining that the recognition of emotions worsens between childhood and adolescence [56]. However, they explain that after twelve weeks of treatment with methylphenidate, an improvement in the recognition of facial emotions, especially negative emotions such as sadness and anger, is identified in children with ADHD, increasing the child’s ability to be empathetic [51].

### 4.3. Empathy

According to this systematic review, empathy was assessed through self-reports and information provided by parents and teachers, where it was identified that children and adolescents, due to the desire for social approval, qualify themselves as empathetic children; however, parents and teachers determine that children with ADHD are less empathetic than the control group, identifying a deficiency in the recognition of anger and sadness in others, which generates a low perception of others’ feelings. The review does not distinguish between cognitive empathy and emotional empathy, considering that there was no separation when applying the instruments. In this dimension, the effects of methylphenidate were evaluated in two studies, confirming that after a certain period of time under pharmacological treatment, the sample managed to improve their empathy levels.

### 4.4. Prosocial Behavior

The review identifies deficits in prosocial behavior as a result of difficulties in social cognition, regardless of the type of ADHD. This dimension was evaluated by teachers and parents, who stated that the type of ADHD determines the severity of these deficits. Children with combined ADHD are more socially aggressive, while children with inattentive ADHD exhibit more passive behaviors; however, neither group, given their deficits, adapts well to the environment or builds social relationships competently, due to their difficulty in generating problem-solving strategies, which leads them to experience social rejection, which may be related to mood difficulties [13]. It is concluded that the types of ADHD interpret emotional and nonverbal signals differently, while their deficits in social functioning are similar; for example, they report that children with combined ADHD have significant difficulties in social performance due to their problematic behaviors, while those with inattentive ADHD base their social difficulty on the lack of attention and perception of the signals in the environment [61].

Adolescents with ADHD exhibit deficiencies in social understanding and problem-solving skills compared to a control group; in addition, less accurate performance is identified when basing their decisions on someone else’s feelings, as well as a difficulty in recognizing sarcasm, showing that the impacts on prosocial behavior affect the quality of life for school-age ADHD patients.

### 4.5. Practical Implications

The results and discussion of this systematic review allow us to initially understand the relationships between social cognition and prosocial behaviors in ADHD, and this understanding allows us to consider the importance of evaluating dimensions such as theory of mind and emotional self-regulation to establish intervention processes that can improve prosocial behaviors at home and at school for children and adolescents with ADHD.

### 4.6. Future Research Lines

It is important to highlight the importance of carrying out studies with longitudinal designs that allow for not only knowing the relationships between dimensions of social cognition and prosocial behaviors, but also generating explanations about the nature of these relationships in ADHD. On the other hand, studies are needed that use not only reported or self-reported measures of the dimensions of social cognition and prosocial behavior, but also experimental tasks, behavioral measurements, and neurophysiological measures, which allow a more objective approach to be taken toward these variables.

## 5. Conclusions

The evidence indicates that children with ADHD have a deficit in social cognition and prosocial behavior, which affects their social interactions and interferes with their communication with others. It is concluded that when the social dimension is affected, it has an impact on other dimensions, showing impacts in terms of quality of life and educational processes. In samples of children with ADHD, the importance of establishing the type of ADHD is identified, in addition to involving their family members or teachers, who are the ones who guide their processes during childhood and adolescence.

The research methodology of this systematic review was based on a quantitative approach, linking cross-sectional and longitudinal studies, and some of these studies were mediated by pharmacological treatment in the participants, making it possible to demonstrate, thanks to the inclusion of control groups, how social cognition and prosocial behavior are affected in children with ADHD, which is a successful outcome for the purpose of this review, highlighting in turn the positive influence of methylphenidate on improvements in the different dimensions evaluated.

The systematic review was performed using research between the years 2000 and 2021, where an increase in the interest of researchers to dive deeper into the subject was identified, thus strengthening the precepts that are developed based on affective neuroscience, expanding the research field not only in social cognition and prosocial behavior, but also increasing the focus on ADHD. These concepts and their relationships are highlighted, thus contributing in theoretical terms to the field of research.

In addition to the main objective of understanding the relationship between social cognition, prosocial behavior, and ADHD, it is also intended to contribute to the field of research influencing the quality of life of children with ADHD, which opens doors to research that becomes part of the educational dimension, assertively building knowledge from a holistic standpoint. The most common limitation of the studies reviewed was the number of participants, which may limit the validity of the studies analyzed. In turn, some of the instruments used are not suitable for the assessment of social cognition, calling into question the reliability and validity of the conclusions.

The cross-sectional studies provide much of the evidence in this review, showing little approach from research with a longitudinal design, which does not allow the understanding of ADHD symptoms or the evolution of deficits in social cognition, so they do not find evidence concerning whether brain maturity or life cycle contribute to the modification of symptoms over time.

Finally, after the effort to integrate the results analyzed in this review, it seems accurate to state that social cognition and prosocial behavior are affected in children and adolescents with ADHD. Although it is unclear whether gender, environment, and type of ADHD influence this deficit, what does seem evident is the social implication that these difficulties cause for the affected participant.

## Figures and Tables

**Figure 1 healthcare-11-01366-f001:**
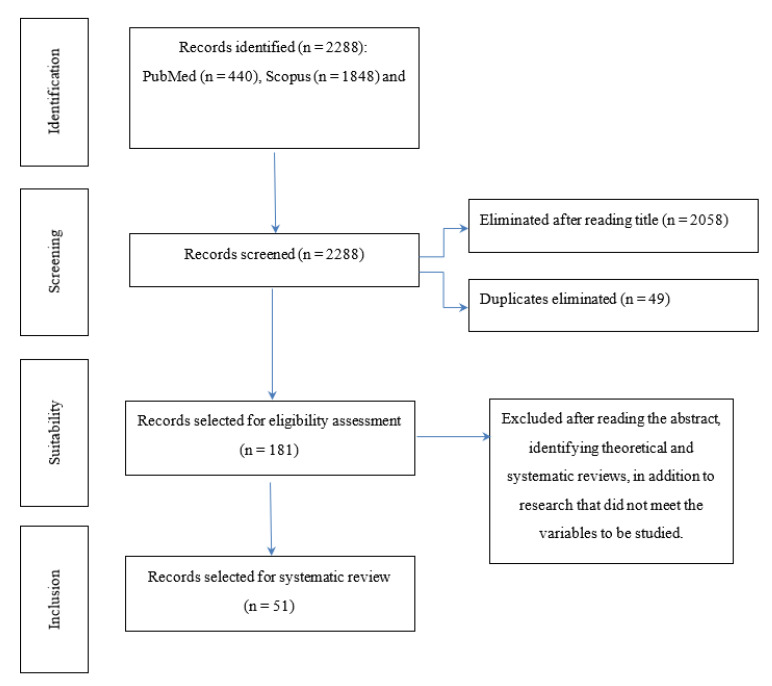
PRISMA flow chart on four levels.

**Table 1 healthcare-11-01366-t001:** Research on social cognition, prosocial behavior, and ADHD.

Study	Sample	Dimensions of Social Cognition/Prosocial Behavior Assessed	Instruments to Measure Social Cognition/Prosocial Behavior	Results
Braaten and Rosen, 2000	United States 29 ADHD and 30 controls Average age of 8.05 years	Self-regulation Empathy Emotion recognition	Behavioral Assessment Scale for Children (BASC; Reynolds and Kampham, 1992). The parent (PRS) and teacher (TRS) versions of BASC. Empathy Task (Ricard and Kamberk-Kilicci (1995). The Empathy Response Task (ERT) (Braaten and Rosen 2000). The Emotion Behavior Checklist (EEC; Izard, Dougherty, Bloxton, and Kotsch, 1974). The Emotional Intensity Scale for Children (EISC) (Braaten and Rosen 2000).	Children with ADHD are less likely to be empathic with the emotions of storybook characters compared to the control group. They have difficulty interpreting and describing the emotions of others and exhibit a more emotional response than the emotions of children without ADHD.
Corbett and Glidden, 2000	United States 37 ADHD and 37 control Ages: 7–12 years	Emotion recognition	Selected slides from the Images of Facial Affect (Ekman, 1976).	Significant differences are evident between the group with ADHD and the control group; children with the disorder also show mild to moderate deficits in their ability to perceive facial expression and prosody.
Maedgen and Carlson, 2000	United States 16 ADHD-C (combined subtype), 14 ADHD-I (inattention subtype), and 17 control Ages: 8–11 years	Emotional regulation Social behavior	Children’s Assertive Behavior Scale (CABS; Michelson and Wood, 1982). Emotional Regulation Task (developed by Saarni (1984)	Both ADHD groups showed social performance deficits in relation to controls.
Charman, Carroll and Sturge, 2001	England 22 ADHD and 22 control Ages: >8 years	Theory of mind Social competence	Happé’s Strange Stories (Happé, 1994).	Control children were rated as more socially competent compared to children with ADHD. In theory of mind, no significant results were identified.
Thurber, Heller and Hinshaw, 2002	United States 49 ADHD and 30 control Ages: 6–12 years	Social behavior	Social goals, interview by Lochman et al. (1993).	Regarding social behavior, the result of girls with ADHD was lower compared to the control group; however, it did not differ significantly.
Jensen and Rosen, 2004	United States 30 ADHD and 37 control Ages: 6–15 years	Emotional regulation	Disruptive Behavior Rating Scale: parent’s version (DBRS-P; Erford, 1993). Emotional Intensity Scale—parent version (Intensity Scale) (Jensen and Rosen 2004). Contingency scale: main version (External Contingency scale) (Jensen and Rosen 2004). Event Scale (Immediate/Future Emotions Scale) (Jensen and Rosen 2004).	The differences in emotional response between children with and without ADHD, although evident in response to positive events in some situations, are stronger in response to negative events.
Pelc et al., 2006	Belgium 30 ADHD and 30 control Ages: 7–12 years	Facial expression recognition	Procedure for decoding facial expressions of emotion.	Children with ADHD exhibited a general deficit in decoding emotional facial expressions, with a specific deficit in identifying anger and sadness.
Yuill and Lyon, 2007	England PHASE 1: 19 ADHD and 19 control PHASE 2: 17 ADHD and 13 control Ages: 5–6 years	Facial expression recognition	Sample photographs for facial emotion recognition (Yuill and Lyon 2007). Emotion matching tasks (Yuill and Lyon 2007).	Children with ADHD have barriers to processing emotional information and difficulties in understanding the links of emotions expressed by others and in identifying the situation in which these emotions occur.
Da Fonseca et al., 2008	France 27 ADHD and 27 control Ages: 5–15 years	Emotion recognition	Schedule for Affective Disorders and Schizophrenia for School-Age Children—Present and Lifetime Version; K-SADS-PL. Conner’s Parent Rating Scale-Revised (CPRS-R). Emotion recognition task (Da Fonseca et al.… 2008).	Children with ADHD present a general deficit in the processing of emotions.
Marton et al., 2008	Canada 50 ADHD and 42 control Ages: 8–12 years	Empathy Social perspective-taking	The Empathy Index for Children and Adolescents (Bryant 1982). “My Child” parent questionnaire that was developed to assess children’s empathy (Kochanska 1992). Interpersonal Negotiation Strategies Interview (INSI; Schultz et al. (1989).	Children with ADHD were rated as less empathic by their parents. Less developed SPT skills than children without ADHD. Social perspective taking is related to IQ and language.
Yang et al., 2009	China 26 ADHD and 30 control	Theory of mind	Appearance-Reality (ToM1). The task was adapted from Flavell et al. Unexpected Location (ToM2). The task was modeled after Sally and Anne’s Task (Baron, Leslie and Fritz, 1985). Unexpected Content (ToM3). This task is adapted from Perner’s et al. know-it-all test.	Children with ADHD did not differ significantly from the control group.
Greenbaum et al., 2009	United States 30 ADHD and 34 control	Emotion processing Theory of mind	Child Behavior Checklist. The Teacher Report Form. The Social Skills Rating Scale. Three subtests of Saltzman-Benaiah and Lalonde’s theory of mind (2007). Four subtests of the Minnesota Test of Affective Processing.	Children with FASD (Fetal Alcohol Spectrum Disorder) present more difficulties in social cognition than the control group. Children with ADHD have difficulties in social cognition (theory of mind and affective processing).
Semrud et al., 2010	United States 76 ADHD-C (combined subtype), 77 ADHD-I (inattention subtype), and 113 control Ages: 9–16 years	Social perception	The Structured Interview for the Diagnostic Assessment of Children (SIDAC) (Puig-Antich and Chambers 1978). Behavioral Assessment System for Children-2 (BASC-2) (Reynolds and Kamphaus 2004). Social Skills Rating System (SSRS). Parent Form (Gresham and Elliott 1990). The Child and Adolescent Social Perception (CASP) measure (Magill-Evans et al. 1996).	The control group showed better behavioral functioning in all areas compared to the clinical groups.
Semrud, 2010	United States 74 ADHD-C (combined subtype) and 78 ADHD-I (inattention subtype) Ages: 7–16 years	Social perception Social functioning	Structured Interview for Diagnostic Assessment (SIDAC) (Puig-Antich and Chambers, 1978). Woodcock-Johnson Battery-III Test of Cognitive Abilities (WJ-ACH III) (Woodcock, McGrew and Mather, 2001). BASC-2 (Reynolds and Kamphaus, 2004). The Child and Adolescent Social Perception Measure (CASP) (Magill-Evans, Koning, Cameron-Sadava, and Manyk, 1996). SSRS-For Parents Form (Gresham and Elliott, 1990).	The ADHD-C (combined subtype) and ADHD-I (inattention subtype) groups showed significant differences in the interpretation of emotional and nonverbal cues in a direct measure of social perception compared to controls.
Sibley, Evans, and Serpell, 2010	United States 27 ADHD and 18 control Ages: 12–18 years	Social competence	Impairment Rating Scale (IRS: Fabiano et al. 2006). Methodology of toys that simulate a situation (Lorch et al. 2000).	The results show that adolescents with ADHD showed deficits in social understanding and problem-solving skills compared to adolescents without ADHD.
Demurie, De Corel, and Roeyers, 2011	Belgium 13 ADHD and 18 control Ages: 11–17 years	Theory of mind Empathy	Interpersonal Reactivity Index (IRI). The Reading the Mind in the Eyes Test (Baron-Cohen et al., 2001). Empathic Accuracy Task.	Participants with ADHD have difficulty reading the thoughts and feelings of others compared to the control group. There were no significant results compared to the control group in the empathic precision test.
García, Jara, and Sánchez, 2011	Spain 18 ADHD caregivers and 26 control caregivers Ages: 8–12 years	Prosocial behavior	Strengths and Difficulties Questionnaire “SDQ-CAS” (Goodman, 2001).	The results show significant differences in the assessment and satisfaction of families of children with ADHD vs. non-ADHD in relation to professional educators. The results of the second test show that teachers perceive significant differences in social skills among children with ADHD and children without ADHD.
Spencer et al., 2011	United States 197 ADHD and 224 control Ages: 6–18 years	Emotional regulation	Child Behavior Checklist (CBCL) (Achencach, T 1997).	The CBCL-DESR profile was associated with higher rates of anxiety and disruptive behavior disorders, evidencing significantly more deficits in emotional and interpersonal functioning in children with ADHD.
Schmitt, Gold, and Rauch, 2012	Germany 21 ADHD and 20 control 10–13 years	Emotion regulation	Questionnaire on Emotion Regulation in Children and Adolescents (FEEL-KJ, Grob and Smolenski, 2005). Parent-rated Strengths and Difficulties Questionnaire (Woerner, Becker and Rothenberger, 2004).	Children with and without ADHD differ in their application of problem-oriented emotional regulation strategies.
Factor, Rosen, and Reyes, 2013	United States 37 ADHD-C and 14 ADHD-I Ages: 8–12 years	Emotional self-awareness Emotional reactivity Emotional dysregulation	The Positive and Negative Affect Schedule—Parent Report (PANAS). PANAS—TA (Phillips, Lonigan, Driscoll and Hooe, 2002) The Emotion Expression Scale for Children (EESC). The EESC (Penza-Clyve and Zeman, 2002). The Reactive-Proactive Aggression Questionnaire (RPAQ). The RPAQ (Raine et al., 2006). Emotion regulation checklist (ERC). The ERC (Shields and Cicchetti, 1997).	Poor emotional awareness was related to reactive behavioral difficulties in children with ADHD.
Caillies et al., 2014	France 15 ADHD and 15 control Ages: 6–12 years	Theory of mind Pragmatic language Understanding irony	The Ice Cream Story (Perner and Wimmer, 1985). The Birthday Story (Tager-Flusberg and Sullivan, 1994). Sixteen ironic stories created by the evaluators.	Children with ADHD have impaired social cognition. The correlation between ToM scores and explanations of ironic comments was highly significant.
Ludlow et al., 2014	England 24 ADHD and 24 control Ages: 12–15 years	Social inference Emotion recognition	Emotion Evaluation Test (EET). Social Inference Test (TA SIT).	Children with ADHD have a deficient recognition of negative and positive emotions.
Maoz et al., 2014	Israel 13 ADHD-I (inattention subtype) and 11 ADHD-C (combined subtype) Ages: 6–12 years	Theory of mind Empathy	Interpersonal Reactivity Index. (IRI) (Davis, 1983) The Faux Pas Recognition Task (FPR), designed by Baron-Cohen et al. The TCT is based on a task (Computerized ToM Task) previously described by Baron-Cohen (1995).	ToM deficits and empathy in children with ADHD- C and ADHD I.
Deschamps, et al., 2014	Netherlands 94 ADHD and 37 control Ages: 6–7 years	Empathy (affective) Prosocial behavior	The Interpersonal Response Task (IRT) (Thurstone and Louis 1990). The parent version of the DISC interview (module E). Child Behavior Checklist (CBCL) (Achencach, T 1997). Griffith Empathy Measurement-Parent Rating (GEM-PR) scales.	Compared to the control group, children with DBD (Disruptive Behavior Disorder with and without ADHD) and those with ADHD alone were rated as less empathic by their teachers, but not by their parents.
Baribeau, et al., 2015	Canada 71 ADHD and 34 control	Social perception	The Reading the Mind in the Eyes Test (Baron-Cohen et al., 2001).	The groups of children with ADHD together with those with ASD (Autism Spectrum Disorder) obtained a lower score in social perception skills.
Mary et al., 2015	Belgium 31 ADHD and 31 control Ages: 8–12 years	Theory of mind	The “Faux Pas” Task (Baron-Cohen et al., 1999). The Reading the Mind in the Eyes Test (Baron-Cohen et al., 2001).	Children with ADHD performed more poorly than control children on ToM.
Mohammadzadeh et al., 2015	Iran 30 ADHD and 30 control	Intentionality (theory of mind)	Moving Shapes Paradigm (Castelli in 2000 and used in Abell et al. (2000) and Castelli et al. (2000).	A deficit in the theory of mind is confirmed in children with ADHD, due to the fact that they showed low performance in the understanding of intentional behaviors.
Orozco and Zuluaga, 2015	Colombia 37 ADHD and 37 control Ages: 6–12 years	Theory of mind (ToM)	Sally, Anne and the Marble Task (Baron, Leslie and Fritz, 1985). Another version of Sally and Anne created by Baron (1989) (Visoverbal). The Ice Cream Van Story (audio).	Children with ADHD performed poorly on false belief tasks, especially first-order ones.
Gumustas et al., 2017	Turkey 65 ADHD and 61 control Ages: 8–14 years	Empathy Emotion recognition	Bryant’s Empathy Index (BEI) (Bryant 1982). Griffith Empathy Measurement-Parent Rating (GEM-PR) scales. The Empathy Response Task (ERT) (Braaten and Rosen 2000). Diagnostic Analysis of Nonverbal Accuracy-2 (DANVA-2) (Nowicki and Duke 1994).	Children with ADHD have similar levels of trait and state empathy and facial expression, but they are still lower than that of control children.
Jusyte, Gulewitschb, and Schönenberg, 2017	Germany 26 ADHD and 27 control Ages: 10–14 years	Emotion recognition	Animated transformation task (Radboud Faces Database (Langner et al., 2010). Software (Version 17.0, Neurobehavioral System, Inc., Berkeley, CA, USA www.neurobs.com accessed on 16 February 2023).	The group effect reflected a lower overall accuracy in all emotional categories in children with ADHD compared to the control group.
Ludlow et al., 2017	England 22 ADHD and 22 control Ages: 11–15 years.	Social inference	The Awareness of Social Inference Test (TASIT, McDonald, Flanagan and Rollins, 2002). British Picture Vocabulary Scale III (BPVS-III, Dunn, Dunn, Styles, and Sewell, 2009).	Children with ADHD demonstrated deficits in understanding paradoxical sarcasm and performed significantly less accurately than children in the control group.
Maoz et al., 2017	Israel 24 ADHD and 36 control Ages: 6–12 years	Theory of mind Empathy	Self-reported empathy questionnaire. Interpersonal Reactivity Index (IRI) (IRI; Davis, 1993). The “Faux Pas” Task (Baron-Cohen et al., 1999).	Boys with ADHD showed significantly lower levels of self-reported empathy on most IRI subscales.
Miranda et al., 2017	Spain 35 ADHD and 30 control Ages: 7–11 years	Theory of mind Social perception	Developmental Neuropsychological Assessment—II (NEPSY-II) (Korkman et al., 2011). The theory of mind inventory (ToMI), Spanish adaptation (Pujals E., Batle S, Camprodon E, Aceña M, Duñó, L 2010).	The ADHD group shows a worse performance in the Verbal ToM task compared to the control group.
Miranda et al., 2017	Spain 35 ADHD and 37 control Ages: 7–11 years	Theory of mind	The theory of mind inventory (ToMI), Spanish adaptation (Pujals E., Batle S, Camprodon E, Aceña M, Duñó, L 2010).	ToM problems were higher in the subgroup of children with ADHD.
Musser and Nigg, 2017	United States 50 ADHD and 50 control Ages: 7–11 years	Emotional dysregulation	Induction (and suppression) Task (Musser et al., 2011). Facial Action Coding System (Ekman, 1992b).	Children with ADHD show weaker consistency between indices of emotion, reactivity, and regulation, including autonomic reactivity and facial affective behavior, compared to children in the control group.
Pitzianti et al., 2017	Italy 23 ADHD and 20 control Ages: 7–15 years.	Theory of mind Emotional regulation	Developmental Neuropsychological Assessment—II (NEPSY-II) (Korkman et al., 2011).	There are no significant differences between the ADHD group and the control group regarding the performance of the ToM tasks as well as regarding the ER (emotional regulation) tasks.
Kara et al., 2017	Turkey 40 ADHD and 11 control Ages: 7–12 years	Emotion recognition	Emotion recognition through facial expression task. Schedule for Affective Disorders and Schizophrenia for School-Age Children—Present and Lifetime Version (K-SADS-PL-Turkish Version). Conner’s Parent Rating Scale-Revised/Long Version (CPRS-R/L).	There is a non-significant difference between the performance of children with pure ADHD and ADHD with comorbid ODD on the facial recognition task.
Elmaghrabi et al., 2018	United States 65 ADHD and 65 control	Emotional regulation	Child Behavior Checklist (CBCL) (Achenbach, T 1997). Eriksen Flanker Task (Castellanos et al., 2005; Di Martino et al., 2008).	The intra-subject variability of the elevated response time (RT-ISV) is related to cognitive rather than emotional dysregulation in children with ADHD.
Basile, Toplak, and Andrade, 2018	Canada 39 ADHD and 42 control Ages: 9–15 years	Emotion recognition	Emotion Recognition Task (ERT) Egger et al., 2011).	The ADHD group was more confident in their recognition of sad and angry faces compared to the control group. The control group and the ADHD group did not differ in emotion recognition accuracy.
Özbaran, Kalyoncu, and Köse, 2018	Turkey 100 ADHD and 100 control Ages: 11–17 years	Theory of mind Emotional regulation/emotional awareness	Difficulties in Emotion Regulation Scale (DERS) (Gratz and Roemer, 2004). The Reading the Mind in the Eyes Test children’s version (Baron and Cohen et al., 2001). The Faces Test (FT. (Ekman, 1972, Ekman and Friesen, 1976)). Unexpected Outcome Test (UOT. Dyck et al., 2001).	There is a significant impact of ToM skills on emotional regulation skills within the ADHD group.
Parke et al., 2018	United States 25 ADHD and 25 control Ages: 7–13 years	Empathy Theory of mind (ToM) Pragmatic language Emotion recognition	Children’s Communication Checklist (CCC-2; Bishop, 2003). The Reading the Mind in the Eyes Test, children’s version (Eyes Test; Baron-Cohen, Wheelwright, Spong, Scahill, and Lawson, 2001) Happé’s Strange Stories (Happé, 1994). The Interpersonal Reactivity Index (IRI; Davis, 1983). Developmental Neuropsychological Assessment, Second Edition (NEPSY-II; Korkman, Kirk, and Kemp, 2007).	Children with ADHD performed significantly worse on measures of cognitive ToM and affect recognition.
Sahin et al., 2018	Turkey 24 ADHD and 24 control Ages: 7–12 years	Theory of mind	Sally, Anne and the Marble Task (Baron, Leslie and Fritz, 1985). Fake Belief Tasks (Chocolate Bar story and Ice Cream Van task). The “Faux Pas” Task (Baron-Cohen et al., 1999). The Reading the Mind in the Eyes Test (Baron-Cohen et al., 2001).	Children with ADHD had affectations in theory of mind and generated difficulties in communication and social interaction.
Waddington et al., 2018	Holland 175 ADHD and 220 control	Emotion recognition	Identification of Facial Emotions (IFE) task of the Amsterdam Neuropsychological Tasks program (ANT; De Sonneville, 1999). The Affective Prosody (AP) task of the Amsterdam Neuropsychological Task program (ANT; De Sonneville, 1999).	Both facial and auditory emotion recognition problems are present in ADHD.
Golubchik, and Weizman, 2019	Israel 25 ADHD with/without MPH Ages: 8–13 years	Theory of mind Socio-emotional functioning	The Reading the Mind in the Eyes Test children’s version (cRMET; Vellante et al. 2013; Vogindroukas et al., 2014). Strengths and Difficulties Questionnaire (SDQ; Goodman, 2001; Mansbach-Kleinfeld et al., 2010; Iizuka et al., 2010). Application before and after MPH.	Poor ToM functioning is associated with lower social-emotional functioning in childrens with ADHD.
Melegari et al., 2019	Italy 86 ADHD and 104 control	Emotional self-regulation	Child Behavior Checklist (CBCL) (Achencach, T 1997). Leiter International Performance Scale: revised (Leiter, 1979). The SDAI and SDAG rating scales (Marzocchi and Cornoldi, 2000).	Children with ADHD show worse behavior when they are involved in group situations with little or no adult supervision.
Tarle et al., 2019	United States 28 ADHD and 35 control Ages: 8–12 years	Emotion regulation	Emotion Regulation Coding (Melnick and Hinshaw, 2000).	Children with ADHD have deficits in emotion regulation compared to control children.
Mohammadzadeh et al., 2019	Iran 30 ADHD and 30 control 7–9 years	Theory of mind	Animated Triangle Test (ATT) (Abell, Happé and Frith (2000) and Castelli et al. (2000).	The ADHD group had a significant ToM impairment relative to the control group.
Dagdelen, 2020	Turkey 60 ADHD and 60 control Ages: 12–16 years	Theory of mind	Schedule for Affective Disorders and Schizophrenia for School-Age Children—Present and Lifetime Version) (K-SADS-PL). The Reading the Mind in the Eyes Test (RMET). The Faux Pas Recognition Test. The hinting task.	ASD children have a greater deficit in social cognition compared to ADHD children. Children with ADHD have deficits in social cognition.
Eyuboglu and Eyuboglu, 2020	Turkey 48 ADHD and 51 control Ages: 12–7 years	Emotional dysregulation	The Difficulties in Emotion Regulation Scale. The Experiences in Close Relationships Scale.	Adolescents with ADHD showed poorer performance in emotion regulation and had higher avoidant attachment.
Yurteri and Şahin, 2020	Turkey 40 ADHD and 40 control Ages: 8–12 years	Theory of mind	The Reading the Mind in the Eyes Test (Baron-Cohen et al., 2001).	We found that serum zonulin levels were significantly higher in the ADHD group compared to the control group.
Singh, Arun, and Bajaj, 2021	India 20 ADHD and 20 control	Theory of mind	Theory of mind inventory (Hutchins et al.… 2011). Theory of mind Task Battery (Hutchins et al.… 2011).	Significant differences are found between Ttheory of mind in children with ADHD and children belonging to the control group.

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
