# Peer review of "Social Cognition and Prosocial Behavior in Children with Attention Deficit Hyperactivity Disorder: A Systematic Review"

_healthcare, 2023, doi:10.3390/healthcare11101366_

Round 1
Reviewer 1 Report
The authors do not make a clear case in the introduction as to why this systematic review is needed and important. The introduction could be strengthened to support the purpose of this review. Additional clarification of constructs (theory of mind, emotional regulation, etc.) would strengthen the paper and help enhance critical review of results. I think the table summarizing the results of the review is too lengthy and provides an unnecessary amount of detail for a table. The conclusions/results are not well-organized for the reader.
Author Response
Thank you for all the comments on our article. These changes have substantially improved the quality of the writing.
Reviewer 2 Report
Thank you for the opportunity to review this paper, which addresses an interesting and important topic: the social cognition deficit of children and adolescents with ADHD. The systematic review approach adopted by the authors is a potentially valuable contribution to knowledge in this area. Nevertheless, it seems to me that there are many points to be reviewed before considering publication and I do not know if the authors will be prepared to carry out this very substantial work.
There are, indeed, important methodological issues to be raised. The most important is that the selection, to appear truly convincing, must involve the intervention of a second co-author and the calculation of an inter-author index. This point is decisive. Without a double quotation, the paper cannot be published. A second important point concerns the selection process itself. The authors start by using 3 databases: Google Scholar, Pub Med and Scopus. Faced with the sheer volume of papers that came up, they set aside Google Scholar, which led them to reject almost 90,000 papers without any argument.
Still on a methodological level, the selection criteria are not sufficiently argued: English, years included, etc. Each choice made must be justified.
Each choice made must be justified.
Another fundamental issue is that the authors report effects on differences in means between two groups (a group of clinical subjects and a group of healthy subjects). It would be useful to have evidence on the proportion of ADHD subjects with empathy or emotion recognition problems. If we really want to learn something about the link between social cognition and ADHD, we need to know whether all children and adolescents with ADHD have problems with social cognition, particularly ToM, and if not, in what proportion.
On the substance, the most important question is the evaluation of this paper in terms of scientific contribution. There is no reference to, or discussion of, previous systematic reviews or meta-analyses, even though the authors state that there are (since these papers were set aside during the selection process). Of course, these papers had to be set aside in this systematic review, but it is essential to justify in the introduction the need for a new systematic review and to return in the discussion to what this systematic review has contributed in relation to previous ones.
At the intersection of substance and form, the paper is written, in general, too imprecisely and in an insufficiently analytical manner, whereas this is the very purpose of a systematic review.
On form:
Page 2, line 69: this is not at all the reason for abandoning Google Scholar. In fact, this reason is never made explicit.
Lines 85-90: What are the criteria for considering the number of papers obtained sufficient for a systematic review?
Page 3, line 100: only papers published in English were selected. This needs an argument!
Line 101: this is not an inclusion criterion.
Figure 1: problem in the lower right square
ADHD-C and ADHD-I: to be defined beforehand
Page 8: FSAD: same
Page 9: Garcia et al (2011): the description of the results is incomprehensible. Why teachers?
Discussion, first paragraph (and the rest): what makes this possible?
How many studies on the link between Theory of Mind & ADHD?
How many of them show a significant effect, of what size? The discussion is too imprecise. Perhaps an intermediate section should be included between the tables summarising the research and the discussion. This would be a sufficiently precise analytical synthesis section to allow an assessment of the systematic nature of the effects and their significance.
Page 26, on longitudinal studies: this was not discussed (or perhaps even specifically presented). Ditto for affective neuroscience.
Lines 453-456: this is too imprecise.
Last line: gender, environment: this is the first time these dimensions are mentioned.
Author Response

(The authors gave the same response as above.)

Reviewer 3 Report
This is a good study with great information. The issues covered are important as we continue to find needs to address ADHD and other related conditions.
I do have a few observations though. In some parts of the study, you relied on lengthy sentences where, at times, a sentence spanned a whole paragraph. At times, these lengthy sentences cause the reader to go back and re-read the sentence... This disrupts the flow so, when possible, I suggest breaking up these lengthy sentences into a few shorter ones.
The inclusion and exclusion criteria sections could be improved by added bullets or numbers.
Finally, table one was very long (16 pages or so). Have you considered breaking it up into multiple tables? This could help the reader a great deal. I would like to add that the discussion that followed was nicely written and well organized.
Good work, overall.
Author Response

(The authors gave the same response as above.)

Reviewer 4 Report
This is an important and comprehensive review. Few thoughts are
Discussion section:elaborate suggestions for additional research,
Methodology and revisions to the manuscript.
Line 21: unclear possibly typo.
Line 44: The initial part of the sentence can be reference separately, as it appears the reference for APA is misleading.
Moher 2009: did you consider using the updated version? Can you speak to the one from 2021 , as the search is 2021. How about network analysis and extensions? Can you comment on that? This may add to your discussion? Would you consider adding Embase as a data search ?
Overall, this highlights an important area when working with ADHD children.
Author Response

(The authors gave the same response as above.)

Round 2
Reviewer 2 Report
The authors have taken most of my comments into account. The manuscript seems to me to be much improved.